# Serum Iron and Risk of Diabetic Retinopathy

**DOI:** 10.3390/nu12082297

**Published:** 2020-07-31

**Authors:** Ying-Jen Chen, Jiann-Torng Chen, Ming-Cheng Tai, Chang-Min Liang, Yuan-Yuei Chen, Wei-Liang Chen

**Affiliations:** 1Department of Ophthalmology, Tri-Service General Hospital, and School of Medicine, National Defense Medical Center, Taipei 114, Taiwan; yj12664@gmail.com (Y.-J.C.); jt66chen@gmail.com (J.-T.C.); mingtai1966@yahoo.com.tw (M.-C.T.); doc30875@yahoo.com.tw (C.-M.L.); 2Department of Pathology, Tri-Service General Hospital Songshan Branch, and School of Medicine, National Defense Medical Center, Taipei 114, Taiwan; fu84fu840618@gmail.com; 3Department of Pathology, Tri-Service General Hospital, and School of Medicine, National Defense Medical Center, Taipei 114, Taiwan; 4Division of Geriatric Medicine, Department of Family and Community Medicine, Tri-Service General Hospital, and School of Medicine, National Defense Medical Center, Taipei 114, Taiwan; 5Division of Environmental Health & Occupational Medicine, Department of Family & Community Medicine, Tri-Service General Hospital, National Defense Medical Center, Taipei 114, Taiwan; 6Department of Biochemistry, National Defense Medical Center, Taipei 114, Taiwan

**Keywords:** serum iron, ferritin, transferrin receptor, diabetic retinopathy

## Abstract

Background: Diabetic retinopathy (DR) is indicated as a major cause of blindness in the world. Emerging evidence supports the interaction of iron metabolism with diabetes. However, little research is available concerning the relationship between iron metabolism and DR. The intent of this paper is to describe the correlation between serum iron and the occurrence of DR. Methods: A total of 5321 participants who underwent related examinations as part of the National Health and Nutrition Examination Survey (2005–2008) were included. DR was defined by the criteria of the Early Treatment for Diabetic Retinopathy Study based on nonmydriatic fundus photography. The cutoff point of serum iron for DR was explored by the receiver operating characteristics curve. The relationship of serum iron with the occurrence of DR was explored by multivariate logistic regression models. Results: Participants with DR had significantly lower serum iron than the control group. Serum iron was negatively correlated with the occurrence of DR after the adjustment of pertinent variables (an odds ratio (OR) of 0.995 (95% CI: 0.992–0.999)). After dividing serum iron into quartiles, the third quartile was associated with DR with an OR of 0.601 (95% CI: 0.418–0.863). Furthermore, the cutoff point of serum iron had an inverse relationship for the occurrence of DR with an OR of 0.766 (95% CI: 0.597–0.984). Conclusion: Serum iron has an inverse association with the occurrence of DR in diabetic adults. The assessment of serum iron levels might be a part of follow-up visits with diabetic patients.

## 1. Introduction

Retinopathy is a well-known microvasculature disease and one of the complications of diabetes and hypertension [1,2]. Diabetic retinopathy (DR) is suggested to be a leading cause of blindness, because its prevalence is approximately 33% in western countries [3,4]. In the 2005–2008 survey years of the National Health and Nutrition Examination Survey (NHANES), data reveal that approximately 30% of people with diabetes mellitus (DM) were at risk of DR [5]. The precipitating factors related to DR have been reported in previous studies [6,7]. Poor control of diabetes and hypertension and long DM duration are strongly associated with DR [8].

Iron plays an important role in catalyzing enzymatic reactions. It is a critical component of the electron transport chain [9]. Iron represented a vital cofactor for guanylate cyclase, an enzyme that synthesizes cGMP, the second messenger in the retinal phototransduction [10,11]. Although an important micronutrient for the physiological processes of several proteins, excess iron can be harmful. Various types of retinal degeneration are suspected to result from iron-mediated retinal cell death [12]. Increased iron accumulation is reported to accelerate the progression of DR in mouse models of DM [13]. The primary focus of this paper was on the correlation of serum iron with DR among these US individuals.

## 2. Methods

### 2.1. Study Population 

The NHANES is executed by the National Center for Health Statistics (NCHS), which contains these US noninstitutionalized civilian participants. All participants underwent comprehensive measurements, such as physical and laboratory examinations and standardized interview questionnaires, including socioeconomic, demographic, and health-related questions. From survey years 2005–2008 of the NHANES, 5321 eligible participants obtained were included in our study. According to the Declaration of Helsinki, this design was approved by the institutional review board of the NCHS. Before examinations, all participants completed informed consents.

### 2.2. Definition of DR and Ophthalmic Examination 

DR was defined by the presence of hemorrhages, hard exudates (HE), cotton wool spots (CWS), microaneurysms (MA), venous beading, intraretinal microvascular abnormalities (IRMA), and retinal new vessels based on the severity scale of the Early Treatment for Diabetic Retinopathy Study (ETDRS). Nonmydriatic fundus photography (TRC-NW6S; Topcon, Tokyo, Japan) was applied for measuring the level of retinopathy in the worse eye. The grades were categorized into no DR, nonproliferative DR, and proliferative DR. Detailed information is listed in the Digital Grading Protocol of the NHANES.

### 2.3. Measurement of Serum Iron, Lead, Cadmium, and Mercury

The serum iron level was measured by the DcX800 method. The system detects the change of ferrous ion absorbance, complexed with the ferro zine iron reagent, at 560 nm at a fixed-time interval. The concentration of iron in the sample is directly proportional to the change in absorbance. The concentration of ferritin was measured by the Roche/Hitachi immunoturbidity assay. The function of a transferrin receptor is to transport iron into cells and maintain iron metabolism. The concentration of a transferrin receptor was also measured by the Roche/Hitachi immunoturbidity assay. The concentration of hemoglobin was determined using the Coulter HMX Hematology Analyzer. The levels of serum lead and cadmium were evaluated by atomic absorption spectrophotometry with Zeeman background correction (SIMAA 6000 model; Perkin-Elmer, Norwalk, CT, USA). Serum inorganic mercury was measured using stannous chloride as a reductant, and we also utilized microwave digestion. 

### 2.4. Covariates Assessment

There are five races/ethnicities in the NHANES database, including Mexican American, non-Hispanic white, non-Hispanic black, other Hispanic, and Others. Current cigarette smoking was defined as having ever smoked and smoking at the time of the survey. Before the determination of biochemical data, blood samples were obtained from participants with 8 h fasting.

### 2.5. Statistical Analysis

The Statistical Package for the Social Sciences (version 18.0, SPSS Inc., Chicago, IL, USA) was applied for analysis. The differences of continuous and categorical variables were investigated using the independent *t*-test and the chi-squared test, respectively. These logistic regression models were used to determine the relationship of serum iron with the presence of DR. Model 1 was unadjusted. Model 2 was adjusted by age, gender, and race. Model 3 = Model 2 and adjusted by alanine aminotransferase (ALT), fasting plasma glucose (FPG), hemoglobin, and cigarette smoking. To assess the optimal cutoff points of serum iron, a receiver operating characteristic (ROC) curve analysis was conducted. Two-sided *p* values of less than 0.05 were considered to indicate significance.

## 3. Results

### 3.1. Association between Lead, Cadmium, Mercury, and the Presence of DR

In Table 1, we show the associations between different kinds of metal in the serum and the presence of DR by a multivariable logistic regression model. Serum iron was negatively associated with the presence of retinopathy with an odds ratio (OR) of 0.995 (95% confidence interval (CI): 0.992–0.998), 0.994 (95% CI: 0.991–0.998), 0.995 (95% CI: 0.992–0.999) in Model 1, Model 2, and Model 3, respectively.

### 3.2. Description of the Study Sample Characteristics

The demographic data of participants with and without DR are shown in Table 2. The average age of participants with DR and without DR was 62.43 and 58.96 years, respectively. Participants with DR had significantly lower levels of serum iron than the comparison group. Variables including age, race/ethnicity, FPG, and ALT had significant differences across groups.

### 3.3. Association between Iron, Ferritin, Transferrin Receptor, and the Presence of DR

In Table 3, the multivariate logistic regression model addressed the association between iron, ferritin, transferrin receptor, and the presence of DR. Serum iron had a substantial correlation with a decreased likelihood of DR with an OR of 0.995 (95% CI: 0.992–0.999) in the fully adjusted model. For the association of ferritin and transferrin receptor with the presence of DR, no significant difference was observed. Moreover, we analyzed this relationship in gender difference and found that serum iron was significantly associated with decreased likelihood of DR, especially in males (data not shown). No significant difference was noted in the relationship between iron, ferritin, transferrin receptor, and the presence of DR in the female population. 

### 3.4. Relationship of Different Quartiles of Iron with the Presence of DR

We divided the levels of serum iron into quartiles and then analyzed the association with the presence of DR (Table 4). the third quartile (Q3) was associated with a decreased likelihood of DR with an OR of 0.583 (95% CI: 0.424–0.802), 0.541 (95% CI: 0.390–0.750), and 0.601 (95% CI: 0.418–0.863) in Model 1, Model 2, and Model 3, respectively. 

### 3.5. Cutoff Point of Iron and DR

Table 5 summarizes the optimal cutoff point of serum iron using a ROC curve analysis. The area under the receiver operating characteristic (AUROC) value was 0.535 (95% CI: 0.512–0.558). The optimal cutoff point was 76.5 with a sensitivity of 56.5% and a specificity of 50%. Furthermore, the relationship between the cutoff point of serum iron and the presence of DR by a multivariate logistic regression model is presented in Table 6. Serum iron was negatively associated with the presence of DR with an OR of 0.737 (95% CI: 0.589–0.923), 0.701 (95% CI: 0.557–0.881), and 0.766 (95% CI: 0.597–0.984) in Model 1, Model 2, and Model 3, respectively.

## 4. Discussion

In the present study, the occurrence of DR had a significant association with serum iron rather than other toxic metals. Participants with DR had lower levels of serum iron than control groups. Compared with the ferritin and transferrin receptors, we highlighted that only serum iron had a significantly inverse relationship with the presence of DR. Notably, the present study was the first to examine the association of serum iron with the presence of DR. 

Our results were contrary to a previous study, which indicated that excessive iron accelerates the progression of DR in mice [13]. Compared with this animal research, we measured the iron level from serum samples rather than directly from retinal samples. Iron overload in the retina was suggested to result in cell death by generating reactive oxygen species and releasing proinflammatory cytokines [14]. Our findings implied that the development of DR might have different potential mechanisms related to serum iron. 

DR is considered a microcirculatory disease of the retina triggered by hyperglycemia. Myriad risk factors have been reported for prediction of the development and severity of DR, including high blood pressure, proteinuria, high serum lipids, the duration of DM, and renal disease. Iron deficiency is suggested to be associated with increased hemoglobin Alc levels in patients with diabetes [15,16]. Ohira et al. proposed that decreased iron levels contributed to elevated gluconeogenic enzymes and serum glucose levels [17]. In a population-based study, lower levels of serum iron were related to glucose dysregulation [18]. In a case-series study of three patients who had DM for an average of 17 years, the background DR of the patients rapidly progressed from a mild-to-moderate grade to a severe proliferative phase, after the development of severe iron deficiency anemia [19]. Consistent with our findings, we found that low serum iron was significantly associated with an increased likelihood of DR. Collectively, iron homeostasis might be a vital step in the progression of DR.

Maintenance of oxygen supply plays a critical role for retinal function [20]. A long duration of hypoxia caused by hyperglycemia or low hemoglobin can lead to retinopathy [21]. Most important of all, iron is incorporated into the heme for transport of oxygen in the blood [22,23]. Nagababu et al. reported that low iron levels enhanced red blood cell oxidative stress [24]. Several studies have described the role of oxidative stress in the development of DR, in which excessive superoxide accelerates the production of advanced glycation end-product (AGE), expression of the AGE receptor, activation of protein kinase C isoforms, and overactivity of the hexosamine pathway [25,26]. Anemia is suggested to accelerate the progression of retinal ischemia in subjects with DR [27]. Andrews et al. also demonstrated that DM patients with anemia exhibited higher expressions of IL-6 and C-reactive protein, compared to diabetic patients without anemia [28]. These inflammatory mediators insulted vascular permeability and functions that harbored a predisposing milieu for DR formation. 

There are some limitations to this study. First, even though a relationship between serum iron and the presence of DR was established, the causal relation could not be addressed due to the study’s cross-sectional design. Second, the NHANES did not include institutionalized citizens; thus, the prevalence of DR might be underestimated. Third, the presence of DM was obtained from a self-reported questionnaire and did not categorize diabetes into Type 1 and Type 2. Finally, we measured serum iron using a single point method; however, the concentration of serum iron varies with time and diurnal variation of serum iron was observed.

## 5. Conclusions

The study investigated the relationship between serum iron and the occurrence of DR. Serum iron had an inverse association with the occurrence of DR in participants with DM after adjusting for confounding variables. Our findings suggested that the evaluation and treatment of serum iron could be a part of follow-up visits in diabetic patients. It is necessary for further studies to explore the mechanism and pathogenesis of DR, in order to design novel therapeutic strategies and improve our knowledge on this issue.

## Figures and Tables

**Table 1 nutrients-12-02297-t001:** Association between iron, lead, cadmium, mercury, and the presence of diabetic retinopathy (DR).

Variables	Model 1 ^a^ OR (95% CI)	*p* Value	Model 2 ^a^ OR (95% CI)	*p* Value	Model 3 ^a^ OR (95% CI)	*p* Value
	*DR*					
**Iron**	0.995 (0.992–0.998)	0.004	0.994 (0.991–0.998)	<0.001	0.995 (0.992–0.999)	0.020
**Lead**	1.037 (0.981–1.096)	0.196	1.008 (0.948–1.071)	0.798	1.023 (0.963–1.086)	0.463
**Cadmium**	0.893 (0.760–1.049)	0.168	0.953 (0.811–1.121)	0.563	0.930 (0.751–1.152)	0.507
**Mercury**	0.955 (0.895–1.020)	0.174	0.951 (0.890–1.017)	0.143	0.977 (0.915–1.043)	0.483

^a^ Adjusted covariates: Model 1: unadjusted; Model 2: Model 1 + age, gender, race/ethnicity; Model 3: Model 2 + glucose, alanine aminotransferase (ALT), hemoglobin, cigarette smoking.

**Table 2 nutrients-12-02297-t002:** Demographic information of study population.

Variables	DR (+)	DR (−)	*p*-Value
**Continuous variables, mean (SD)**			
Age (years)	62.43 (11.79)	58.96 (12.42)	<0.001
Serum Iron (ug/dL)	81.34 (31.20)	85.76 (35.43)	0.008
Ferritin (ng/dL)	57.09 (70.56)	62.92 (70.16)	0.717
Transferrin Receptor (mg/L)	4.74 (2.76)	4.00 (2.38)	0.195
Fasting Plasma Glucose (mg/dL)	135.83 (71.79)	102.67 (31.91)	<0.001
ALT (U/L)	26.19 (34.73)	25.53 (16.95)	0.004
Hemoglobin (g/dL)	14.07 (1.68)	14.29 (1.52)	<0.001
**Category Variables, (%)**			
Gender (male)	389 (14.7)	2257 (85.3)	0.002
Mexican American	127 (15.3)	701 (84.7)	<0.001
Other Hispanic	54 (14.6)	316 (85.4)	
Non-Hispanic White	298 (10.4)	2560 (89.6)	
Non-Hispanic Black	211 (19.4)	878 (80.6)	
Others	20 (11.4)	156 (88.6)	
Cigarette smoking	120 (12.8)	817 (87.2)	0.853

**Table 3 nutrients-12-02297-t003:** Association between iron, ferritin, transferrin receptor, and the presence of DR.

Variables	Model 1 ^a^ OR (95% CI)	*p* Value	Model 2 ^a^ OR (95% CI)	*p* Value	Model 3 ^a^ OR (95% CI)	*p* Value
	*DR*					
**Iron**	0.995 (0.992–0.998)	0.004	0.994 (0.991–0.998)	<0.001	0.995 (0.992–0.999)	0.020
**Ferritin**	1.000 (0.994–1.006)	0.995	1.000 (0.994–1.006)	0.998	0.998 (0.991–1.005)	0.524
**Transferrin Receptor**	0.988 (0.805–1.214)	0.909	0.988 (0.803–1.215)	0.908	1.052 (0.780–1.417)	0.741

^a^ Adjusted covariates: Model 1: unadjusted; Model 2: Model 1 + age, gender, race/ethnicity; Model 3: Model 2 + glucose, alanine aminotransferase (ALT), hemoglobin, cigarette smoking

**Table 4 nutrients-12-02297-t004:** Relationship of different quartiles of iron with DR.

	Variables	Model 1 ^a^ OR (95% CI)	*p* Value	Model 2 ^a^ OR (95% CI)	*p* Value	Model 3 ^a^ OR (95% CI)	*p* Value
		*DR*					
**Iron**	**Q1 vs. Q4**	0.796 (0.586–1.082)	0.145	0.753 (0.553–1.027)	0.074	0.790 (0.568–1.100)	0.163
	**Q2 vs. Q4**	0.853 (0.627–1.161)	0.312	0.795 (0.582–1.086)	0.149	0.833 (0.593–1.170)	0.293
	**Q3 vs. Q4**	0.583 (0.424–0.802)	<0.001	0.541 (0.390–0.750)	<0.001	0.601 (0.418–0.863)	0.006

^a^ Adjusted covariates: Model 1: unadjusted; Model 2: Model 1 + age, gender, race/ethnicity; Model 3: Model 2 + FPG, ALT, hemoglobin, cigarette smoking.

**Table 5 nutrients-12-02297-t005:** Optimal cutoff points of serum iron.

AUC (95%CI)	0.535 (0.512–0.558)
Sensitivity	56.5%
Specificity	50%
*p*-value	<0.001
Cutoff value	76.5

**Table 6 nutrients-12-02297-t006:** Association between cutoff points of iron and the presence of DR.

Variables	Model 1 ^a^ OR (95% CI)	*p* Value	Model 2 ^a^ OR (95% CI)	*p* Value	Model 3 ^a^ OR (95% CI)	*p* Value
	*DR*					
**Cutoff Points of Iron (76.5 ug/dL)**	0.737 (0.589–0.923)	0.008	0.701 (0.557–0.881)	0.002	0.766 (0.597–0.984)	0.037

^a^ Adjusted covariates: Model 1: unadjusted; Model 2: Model 1 + age, gender, race/ethnicity; Model 3: Model 2 + FPG, ALT, hemoglobin, cigarette smoking.

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
