# Peer review of "Serum Iron and Risk of Diabetic Retinopathy"

_nutrients, 2020, doi:10.3390/nu12082297_

Round 1
Reviewer 1 Report
The work analyses data of 5321 participants to the National Health and Nutrition Examination Survey for relationships between diabetic retinopathy and metals in serum. A negative correlation between serum iron and retinopathy was found. However no significant difference between the presence and absence of retinopathy was found in other indices of iron status. It is suggested to monitor diabetic patients for iron deficiency anemia. The work has some major problems.
- Table 2 shows most male subjects were retinopathy negative (85.3%) and only a few were positive (14.7%). This implies that female subjects were mostly retinopathy positive, and females are known to have lower serum iron levels, particularly in the fertile period of life. Thus, the values of the two genders should be analyzed separately, and it should be verified if there is a gender effect in the overall data of serum iron.
-The title is not correct, since the work does not deal with iron metabolism, but only with serum iron and does not give any indication that the iron status is altered in diabetic retinopathy. In fact the reduced serum iron if not due to a gender effect may also be related to an inflammatory response.
- The discussion is centered on the importance of anemia in diabetes and in retinopathy, but the work do not show any evidence of anemia in the subjects analyzed, the mean values being in the normal range. Thus the discussion should focus on serum iron.
Author Response
July 25, 2020
Dear editor:
Thank you for your encouraging letter concerning our manuscript entitled “Iron Metabolism and Risk of Diabetic Retinopathy” by Ying-Jen Chen et al, for publication in Article of Nutrients.
We are extremely grateful to you and the reviewers for the constructive critique of our manuscript. We have responded to each of the comments of the referees on separate sheets and deeply appreciated your suggestions that have led to a significant improvement in this article. In response to your comments, we have revised the manuscript to enhance article readability. Several new sections of text are added. We have also reedited the abstract, introduction, results, methods, and discussion sections. All the changes are labeled in red color. Accordingly, we resubmit this article to Nutrients.
We look forward to your prompt reply.
Yours sincerely,
Wei-Liang Chen, M.D. Ph.D.
Division of Geriatric Medicine, Department of Family Medicine, Tri-Service General Hospital, National Defense Medical Center,
Number 325, Section 2, Chang-gong Rd, Nei-Hu District, 114, Taipei, Taiwan, ROC.
Tel: +886-2-87923311 ext. 16567
Fax: +886-2-87927057
E-mail: weiliang0508@gmail.com
Answer to Editor’s and Reviewer's comments
Thank you for your positive comments on this manuscript. The responses to the raised questions are below.
Reviewer: 1
The work analyses data of 5321 participants to the National Health and Nutrition Examination Survey for relationships between diabetic retinopathy and metals in serum. A negative correlation between serum iron and retinopathy was found. However no significant difference between the presence and absence of retinopathy was found in other indices of iron status. It is suggested to monitor diabetic patients for iron deficiency anemia. The work has some major problems.
- Table 2 shows most male subjects were retinopathy negative (85.3%) and only a few were positive (14.7%). This implies that female subjects were mostly retinopathy positive, and females are known to have lower serum iron levels, particularly in the fertile period of life. Thus, the values of the two genders should be analyzed separately, and it should be verified if there is a gender effect in the overall data of serum iron.
Response: Thank you for your thorough review and salient observations. We have revised the sentence based on your recommendation.
Moreover, we analyzed this relationship in gender difference that serum iron was significantly associated with decreased likelihood of DR, especially in males (data not shown). No significant difference was noted in the relationship between iron, ferritin, transferrin receptor, and the presence of DR in female population. (page 6, line 132-135)
- The title is not correct, since the work does not deal with iron metabolism, but only with serum iron and does not give any indication that the iron status is altered in diabetic retinopathy. In fact the reduced serum iron if not due to a gender effect may also be related to an inflammatory response.
Response: Thank you for your thorough review and salient observations. We have revised the title of the article based on your recommendation.
Serum Iron and the Risk of Diabetic Retinopathy (page 1, line 2)
- The discussion is centered on the importance of anemia in diabetes and in retinopathy, but the work do not show any evidence of anemia in the subjects analyzed, the mean values being in the normal range. Thus the discussion should focus on serum iron.
Response: Thank you for your thorough review and salient observations. We have revised the Discussion section based on your recommendation.
DR is considered a microcirculatory disease of the retina triggered by hyperglycemia. Myriads of risk factors have been reported for prediction of the development and severity of DR, including high blood pressure, proteinuria, high serum lipids, the duration of DM, and renal disease. Iron deficiency is suggested to be associated with increased hemoglobin Alc levels in patients with diabetes1,2. Ohira et al. proposed that iron deficiency contributed to upregulated expression of genes encoding gluconeogenic enzymes and increased serum glucose levels3. In a population-based study, lower levels of serum iron was related to glucose dysregulation4. In a case-series studies of three patients who had DM for an average of 17 years, the background DR of the patients rapidly progressed from mild-to-moderate grade to a severe proliferative phase after the development of severe iron deficiency anemia5. Consistent with our findings, we found that low serum iron was significantly associated with an increased likelihood of DR. Collectively, iron homeostasis might be a vital step in the progression of DR. (page 10, line 177-187)
Reference:
- Christy, A.L.; Manjrekar, P.A.; Babu, R.P.; Hegde, A.; Rukmini, M.S. Influence of iron deficiency anemia on hemoglobin A1c levels in diabetic individuals with controlled plasma glucose levels. Iranian biomedical journal 2014, 18, 88-93, doi:10.6091/ibj.1257.2014.
- Coban, E.; Ozdogan, M.; Timuragaoglu, A. Effect of iron deficiency anemia on the levels of hemoglobin A1c in nondiabetic patients. Acta haematologica 2004, 112, 126-128, doi:10.1159/000079722.
- Ohira, Y.; Chen, C.S.; Hegenauer, J.; Saltman, P. Adaptations of lactate metabolism in iron-deficient rats. Proceedings of the Society for Experimental Biology and Medicine. Society for Experimental Biology and Medicine (New York, N.Y.) 1983, 173, 213-216, doi:10.3181/00379727-173-41633.
- Krisai, P.; Leib, S.; Aeschbacher, S.; Kofler, T.; Assadian, M.; Maseli, A.; Todd, J.;Estis, J.; Risch, M.; Risch, L., et al. Relationships of iron metabolism with insulin resistance and glucose levels in young and healthy adults. European Journal of Internal Medicine 2016, 32, 31-37, doi:https://doi.org/10.1016/j.ejim.2016.03.017.
- Shorb, S.R. Anemia and diabetic retinopathy. American journal of ophthalmology 1985, 100, 434-436.
Last, we are deeply honored by the time and effort you spent in reviewing this manuscript. In reviewing and revising our text, we are motivated to read more and thus learn more from your criticisms.

Reviewer 2 Report
General comments:
Diabetic retinopathy (DR) is the primary microvasculature disease of diabetic patients, and the vital factor to cause blindness in western countries. The main purpose of this study is to determine the association between serum iron and DR in individuals obtained from the National Health and Nutrition Examination 64 Survey (NHANES) from 2005 to 2008. The main finding of this study includes that the serum iron levels had a significant inverse relationship with the presence of DR. This is a well-written manuscript; however, there are still some major and minor concerns needed to be addressed.
Comments to the authors:
- In the previous publication (Adele Bahar et al., Caspian J Intern Med 2013; 4(4): 759-762), it indicates that anemia is defined as hemoglobin level less than 13 g/dl in men and 12 g/dl in women. According to this definition, the participants in this study do not have anemia (Table 1). If this is the case, how can the authors discuss iron deficiency anemia contributing to DR?
- In this manuscript, the authors divided the participants into Model 1, and Model 2, and Model 3. However, they do not provide a clear description on how they select Model 2 and Model 3. For example, the participants include Mexican American, Other Hispanic, Non-Hispanic White, and Non-Hispanic Black (Table 1). Thus, the key question is: Does the association between the serum iron levels and DR found in all races or a particular race.
- It is interesting to show that the serum iron levels had a significant inverse relationship with the presence of DR. Is there any study in the literature that iron supplement for anemia has beneficial effects in DR or cardiovascular diseases?
- In Table 2, the parameters in the variable column are not well arranged.
- The authors should change “Table 1. Association between lead, cadmium, mercury, and the presence of DR” to “Table 1. Association between iron, lead, cadmium, mercury, and the presence of DR.”
Author Response
July 25, 2019
Dear editor:
Thank you for your encouraging letter concerning our manuscript entitled “Iron Metabolism and Risk of Diabetic Retinopathy” by Ying-Jen Chen et al, for publication in Article of Nutrients.
We are extremely grateful to you and the reviewers for the constructive critique of our manuscript. We have responded to each of the comments of the referees on separate sheets and deeply appreciated your suggestions that have led to a significant improvement in this article. In response to your comments, we have revised the manuscript to enhance article readability. Several new sections of text are added. We have also reedited the abstract, introduction, results, methods, and discussion sections. All the changes are labeled in red color. Accordingly, we resubmit this article to Nutrients.
We look forward to your prompt reply.
Yours sincerely,
Wei-Liang Chen, M.D. Ph.D.
Division of Geriatric Medicine, Department of Family Medicine, Tri-Service General Hospital, National Defense Medical Center,
Number 325, Section 2, Chang-gong Rd, Nei-Hu District, 114, Taipei, Taiwan, ROC.
Tel: +886-2-87923311 ext. 16567
Fax: +886-2-87927057
E-mail: weiliang0508@gmail.com
Answer to Editor’s and Reviewer's comments
Thank you for your positive comments on this manuscript. The responses to the raised questions are below.
Reviewer 2:
General comments
Diabetic retinopathy (DR) is the primary microvasculature disease of diabetic patients, and the vital factor to cause blindness in western countries. The main purpose of this study is to determine the association between serum iron and DR in individuals obtained from the National Health and Nutrition Examination 64 Survey (NHANES) from 2005 to 2008. The main finding of this study includes that the serum iron levels had a significant inverse relationship with the presence of DR. This is a well-written manuscript; however, there are still some major and minor concerns needed to be addressed.
Comments to the authors:
- In the previous publication (Adele Bahar et al., Caspian J Intern Med 2013; 4(4): 759-762), it indicates that anemia is defined as hemoglobin level less than 13 g/dl in men and 12 g/dl in women. According to this definition, the participants in this study do not have anemia (Table 1). If this is the case, how can the authors discuss iron deficiency anemia contributing to DR.
Response: Thank you for your thorough review and salient observations. We have revised the Discussion section based on your recommendation.
DR is considered a microcirculatory disease of the retina triggered by hyperglycemia. Myriads of risk factors have been reported for prediction of the development and severity of DR, including high blood pressure, proteinuria, high serum lipids, the duration of DM, and renal disease. Iron deficiency is suggested to be associated with increased hemoglobin Alc levels in patients with diabetes1,2. Ohira et al. proposed that iron deficiency contributed to upregulated expression of genes encoding gluconeogenic enzymes and increased serum glucose levels3. In a population-based study, lower levels of serum iron was related to glucose dysregulation4. In a case-series studies of three patients who had DM for an average of 17 years, the background DR of the patients rapidly progressed from mild-to-moderate grade to a severe proliferative phase after the development of severe iron deficiency anemia5. Consistent with our findings, we found that low serum iron was significantly associated with an increased likelihood of DR. Collectively, iron homeostasis might be a vital step in the progression of DR. (page 10, line 177-187)
Reference:
- Christy, A.L.; Manjrekar, P.A.; Babu, R.P.; Hegde, A.; Rukmini, M.S. Influence of iron deficiency anemia on hemoglobin A1c levels in diabetic individuals with controlled plasma glucose levels. Iranian biomedical journal 2014, 18, 88-93, doi:10.6091/ibj.1257.2014.
- Coban, E.; Ozdogan, M.; Timuragaoglu, A. Effect of iron deficiency anemia on the levels of hemoglobin A1c in nondiabetic patients. Acta haematologica 2004, 112, 126-128, doi:10.1159/000079722.
- Ohira, Y.; Chen, C.S.; Hegenauer, J.; Saltman, P. Adaptations of lactate metabolism in iron-deficient rats. Proceedings of the Society for Experimental Biology and Medicine. Society for Experimental Biology and Medicine (New York, N.Y.) 1983, 173, 213-216, doi:10.3181/00379727-173-41633.
- Krisai, P.; Leib, S.; Aeschbacher, S.; Kofler, T.; Assadian, M.; Maseli, A.; Todd, J.;Estis, J.; Risch, M.; Risch, L., et al. Relationships of iron metabolism with insulin resistance and glucose levels in young and healthy adults. European Journal of Internal Medicine 2016, 32, 31-37, doi:https://doi.org/10.1016/j.ejim.2016.03.017.
- Shorb, S.R. Anemia and diabetic retinopathy. American journal of ophthalmology 1985, 100, 434-436.
- In this manuscript, the authors divided the participants into Model 1, and Model 2, and Model 3. However, they do not provide a clear description on how they select Model 2 and Model 3. For example, the participants include Mexican American, Other Hispanic, Non-Hispanic White, and Non-Hispanic Black (Table 1). Thus, the key question is: Does the association between the serum iron levels and DR found in all races or a particular race.
Response: Thank you for your thorough review and salient observations. According to published literatures, influential demographic factors and clinical standpoints could be used for covariate adjustments. Old age, female gender, cigarette smoking, and impaired liver function are reported to be associated with risk of diabetic retinopathy1-4. The prevalence of diabetic retinopathy varies from different ethnicities5. Anemia is regarded as a risk factor for diabetic retinopathy6. As result, we included these factors as covariates for adjusting the relationship between serum iron and diabetic retinopathy.
Reference:
- Fong, D. S. et al. Diabetic Retinopathy. 26, s99-s102, Diabetes Care (2003).
- Kajiwara, A. et al. Gender differences in the incidence and progression of diabetic retinopathy among Japanese patients with type 2 diabetes mellitus: a clinic-based retrospective longitudinal study. Diabetes research and clinical practice 103, e7-10, doi:10.1016/j.diabres.2013.12.043 (2014).
- Moss, S. E., Klein, R. & Klein, B. E. Association of Cigarette Smoking With Diabetic Retinopathy. 14, 119-126, doi:10.2337/diacare.14.2.119 %J Diabetes Care (1991).
- Lin, T.-Y., Chen, Y.-J., Chen, W.-L. & Peng, T.-C. The Relationship between Nonalcoholic Fatty Liver Disease and Retinopathy in NHANES III. PLoS One 11, e0165970-e0165970, doi:10.1371/journal.pone.0165970 (2016).
- Sivaprasad, S., Gupta, B., Crosby-Nwaobi, R. & Evans, J. Prevalence of diabetic retinopathy in various ethnic groups: a worldwide perspective. Survey of ophthalmology 57, 347-370, doi:10.1016/j.survophthal.2012.01.004 (2012).
- Ranil, P. K. et al. Anemia and diabetic retinopathy in type 2 diabetes mellitus. The Journal of the Association of Physicians of India 58, 91-94 (2010).
3. It is interesting to show that the serum iron levels had a significant inverse relationship with the presence of DR. Is there any study in the literature that iron supplement for anemia has beneficial effects in DR or cardiovascular diseases?
Response: Thank you for your thorough review and salient observations. After thoroughly searching for related literatures, we didn’t find any paper about the beneficial effect of iron supplement on DR or cardiovascular diseases.
- In Table 2, the parameters in the variable column are not well arranged.
Response: Thank you for your thorough review and salient observations. We have revised the Table 2 based on your recommendation.
- The authors should change “Table 1. Association between lead, cadmium, mercury, and the presence of DR” to “Table 1. Association between iron, lead, cadmium, mercury, and the presence of DR.”
Response: Thank you for your thorough review and salient observations. We have changed the title of Table 1 based on your recommendation.
Table 1. Association between iron, lead, cadmium, mercury, and the presence of DR (page 5, line 112)
Last, we are deeply honored by the time and effort you spent in reviewing this manuscript. In reviewing and revising our text, we are motivated to read more and thus learn more from your criticisms.

Round 2
Reviewer 1 Report
The answers to the points I raised are satisfactory
Reviewer 2 Report
The authors did a good job to address my comments. No further comment.